# The Molecular Evolution, Structure, and Function of Coproporphyrinogen Oxidase and Protoporphyrinogen Oxidase in Prokaryotes

**DOI:** 10.3390/biology12121527

**Published:** 2023-12-15

**Authors:** Marcel Zámocký, Stefan Hofbauer, Thomas Gabler, Paul G. Furtmüller

**Affiliations:** 1Laboratory of Phylogenomic Ecology, Institute of Molecular Biology, Slovak Academy of Sciences, Dúbravská Cesta 21, SK-84551 Bratislava, Slovakia; marcel.zamocky@savba.sk; 2Department of Inorganic Chemistry, Faculty of Natural Sciences, Comenius University in Bratislava, Mlynská Dolina, Ilkovičova 6, SK-84215 Bratislava, Slovakia; 3Institute of Biochemistry, Department of Chemistry, University of Natural Resources and Life Sciences, Vienna, Muthgasse 18, A-1190 Vienna, Austria; stefan.hofbauer@boku.ac.at (S.H.); thomas.gabler@boku.ac.at (T.G.)

**Keywords:** phylogenetic analysis, sequence alignment, prokaryotic heme biosynthesis, coproporphyrin-dependent pathway, protoporphyrin-dependent pathway, coproporphyrinogen III oxidase, protoporphyrinogen IX oxidase

## Abstract

**Simple Summary:**

Recent research has revealed that heme biosynthesis can occur in three different pathways, the long-established “classical pathway” or protoporphyrin-dependent pathway, the recently discovered coproporphyrin-dependent pathway, and the siroheme-dependent pathway. Two oxidases of the classical and the coproporphyrinogen-dependent pathway that catalyze similar reactions were analyzed for their phylogenetic origin. The results showed that oxidases from the classical pathways originate from cyanobacteria, whereas oxidases from the coproporphyrinogen-dependent pathway originate from monoderm *Actinomycetota* and *Bacillota*. The structural similarities and differences between these two oxidases are discussed based on their protein sequence alignment and a structural comparison.

**Abstract:**

Coproporphyrinogen oxidase (CgoX) and protoporphyrinogen oxidase (PgoX) catalyze the oxidation of the flexible cyclic tetrapyrrole of porphyrinogen compounds into fully conjugated, planar macrocyclic porphyrin compounds during heme biosynthesis. These enzymes are activated via different pathways. CgoX oxidizes coproporphyrinogen III to coproporphyrin III in the coproporphyrin-dependent pathway, whereas PgoX oxidizes protoporphyrinogen IX to protoporphyrin IX in the penultimate step of the protoporphyrin-dependent pathway. The phylogenetic analysis presented herein demonstrates a clear differentiation between the two enzyme classes, as evidenced by the clustering of sequences in distinct clades, and it shows that, at the origin of porphyrinogen-type oxidase evolution, PgoXs from cyanobacteria were found, which were noticeably separated from descendant PgoX representatives of *Deltaproteobacteria* and all later PgoX variants, leading to many eukaryotic clades. CgoX sequences originating from the monoderm *Actinomycetota* and *Bacillota* were well separated from the predecessor clades containing PgoX types and represent a peculiar type of gene speciation. The structural similarities and differences between these two oxidases are discussed based on their protein sequence alignment and a structural comparison.

## 1. Introduction

Heme (iron protoporphyrin IX) is found in the vast majority of the known forms of life, where it plays an essential role as a cofactor for a wide variety of proteins involved in many metabolic and respiratory processes, such as electron transport, signaling, catalysis, transcription, and gas sensing. Heme is also found in and required by organisms that do not synthesize it. Many of them are able to take up heme from the environment, their food, or their host. For example, *C. elegans* and some parasitic worms are heme auxotrophs [1] or have an incomplete heme biosynthetic pathway, such as *Enterococcus faecalis*, *Dehalococcoides*, and *Thermotoga* [2,3,4]. Nevertheless, a complete pathway for heme biosynthesis exists in almost all organisms that use heme. The ancient core pathway, which involves 5-aminolevulinic acid (5-ALA) as the starting intermediate and ends with uroporphyrinogen III as the end product, is highly conserved. Uroporphyrinogen III serves as a precursor molecule for three different pathways involved in heme synthesis. These are (i) the long-established “classical pathway” or protoporphyrin-dependent (PPD) pathway, which is found in Gram-negative bacteria and all eukaryotes; (ii) the recently discovered coproporphyrin-dependent (CPD) pathway [5], which is found only in Gram-positive bacteria, with a few exceptions; and (iii) the siroheme-dependent (SHD) pathway, which is the oldest but least common of the three pathways. The PPD and CPD biosynthetic pathways involve the decarboxylation of uroporphyrinogen III to coproporphyrinogen III; however, they differ in the sequence of the three enzymatic reactions converting coproporphyrinogen III to protoporphyrin IX or heme *b* (Figure 1A). 

In the PPD pathway, coproporphyrinogen III is first oxidatively decarboxylated by coproporphyrinogen decarboxylase (CgdC)/coproporphyrinogen dehydrogenase (CgdH) to protoporphyrinogen IX, followed by the oxidation of this intermediate to protoporphyrin IX by protoporphyrinogen IX oxidase (PgoX), and finally, iron is incorporated by the protoporphyrin ferrochelatase (PpfC) to form the final product heme *b* (Figure 1) [4,6,7]. In the CPD pathway, coproporphyrinogen III is oxidized to coproporphyrin III by coproporphyrinogen oxidase (CgoX), followed by the incorporation of iron by coproporphyrin ferrochelatase (CpfC) yielding the four-propionate iron-coproporphyrin III (coproheme), and then the decarboxylation, by coproheme decarboxylase (ChdC), of two propionate groups at positions 2 and 4 of coproheme to yield protoheme, which has vinyl substituents at those respective positions (Figure 1) [4,5].

The discovery of the CPD pathway in Gram-positive bacteria revealed an evolutionary transition between the siroheme-dependent pathway first found in Archaea and the PPD pathway found in Gram-negative bacteria and eukaryotes. The evolutionary significance of this pathway is obvious because chlorophyll synthesis requires protoporphyrin IX. Therefore, the evolution of chlorophyll-based photosynthesis may not have occurred before that of the PPD pathway [8]. 

However, all studies on the enzymes involved in the CPD pathway that were performed prior to the discovery of this pathway [5] were performed in the belief that these enzymes are part of the PPD pathway; therefore, protoporphyrinogen IX and protoporphyrin IX (or other similar porphyrins with two vinyl and two propionate groups) were used as substrate instead of coproporphyrinogen III and coproporphyrin III. While these studies have provided valuable information on the overall structure, iron-binding site, or protonation site of porphyrin, with knowledge of the CPD pathway the investigation of essential steps, such as substrate binding, conversion, release, and regulation with the endogenous substrate, has started [9,10,11,12].

In this work, we focused on two enzymes that are at different positions in each pathway, but are responsible for the oxidation of the flexible, cyclic tetrapyrrole of the porphyrinogen compounds into the fully conjugated, planar macrocyclic porphyrin compounds (Figure 1B). This study will be an important starting point to initiate systematic investigations on coproporphyrinogen III oxidases. Currently, a very limited amount of experimental data on a few representatives of this enzyme class is available. Therefore, a robust structural and phylogenetic analysis, which is provided in this work, will act as a solid base for future studies. Protoporphyrinogen IX oxidase catalyzes the oxidation of protoporphyrinogen IX to protoporphyrin IX (PgoX), whereas coproporphyrinogen III oxidase (CgoX) converts coproporphyrinogen III to coproporphyrin III. In most eukaryotes studied, PgoX have been shown to be mitochondrial membrane-associated proteins, supposedly due to a hydrophobic conserved region [13]. In plants there exist two isoforms, one of which is located in the envelope membrane of chloroplasts while the other is situated on the outer surface of the inner mitochondrial membrane. But it is unclear whether PgoX is inserted in to the membrane as a dimer or monomer; both insertions are structurally possible [14]. However, CgoX is a soluble monomer, as has been demonstrated in studies with *B. subtilis*. Both proteins are flavin adenine dinucleotide (FAD)-containing enzymes with monomer sizes of approximately 50–55 kDa that catalyze the six-electron oxidation of porphyrinogen compounds. Three dioxygen molecules accept six hydrogen atoms from the porphyrinogen compounds, forming three molecules of hydrogen peroxide (Figure 1B). Since there is only one FAD group per enzyme/monomer, it is evident that the complete six-electron oxidation must proceed via three two-electron steps with a tetrahydro and dihydro intermediate. This reaction mechanism has been proposed by Koch et al. for PgoX from common tobacco (*Nicotiana tabacum*) [14]. Koch et al. modelled the binding of protoporphyrinogen IX and protoporphyrin IX. In this model, the negatively charged propionyl groups are orientated towards the solvent-exposed parts of the active site. The methylene bridge between rings A and D of the modelled protoporphyrinogen IX is aligned with the reactive N5 atom of the FAD, which leads to the assumption that this position is initially oxidized by the FAD. Further oxidation can only take place from this position, as the narrow cavity of the active center only allows protoporphyrinogen IX to bind in this orientation and excludes rotations of the substrate. By hydrogen rearrangements through imine–enamine tautomerizations, all hydride abstraction reactions occur from the methylene bridge between rings A and D, whereas the reaction mechanism for the conversion of coproporphyrinogen III to coproporphyrin III in the CPD pathway remains to be elucidated. 

In this study, we have reconstructed the molecular evolution of CgoX and PgoX, which oxidize cyclic tetrapyrrole compounds into fully conjugated planar macrocyclic porphyrin compounds. We present the phylogenetic relationships of the corresponding protein family, introduce their representative sequence signatures and essential amino acids at cofactor- and substrate-binding sites, and critically discuss their structural differences from the known, resolved 3D structures.

## 2. Methods

### 2.1. List of Used Protein Sequences and Their Source

A selection of 74 full-length CgoX and PgoX protein sequences were collected from public databases (UniProt, NCBI). BLAST searches were performed on both sequence databases to collect a representative set of oxidases from both pathways. All sequences used in this study were verified for the presence of porphyrinogen oxidases’ motifs (Appendix A).

### 2.2. Multiple Protein Sequence Alignments

Multiple sequence alignment of the 74 selected porphyrin oxidase protein sequences for phylogenetic reconstruction were performed using the Clustal-X program implemented in the MEGA-X package with up to 1000 iterations [15]. Pairwise alignment was performed with a gap opening penalty of 10 and a gap extension penalty of 0.1, whereas multiple alignment was performed with a gap opening penalty of 10 and a gap extension penalty of 0.2. The aligned dataset of protein sequences was exported as comma-separated values (.csv) and imported into an Excel spreadsheet for formatting and representation.

### 2.3. Reconstruction of Evolutionary Relationships

Phylogenetic relationships of the 74 aligned protein sequences were analyzed using the MEGA-X suite [16]. The maximum likelihood (ML) method of phylogenetic reconstruction was chosen with the application of the Le Gascuel model of amino acid substitutions [17] as the statistically proven model (obtained in MEGA-X) with the lowest Bayesian Information Criterion score among the 56 analyzed models. The following optimized parameters were determined and used for the final reconstruction: 1000 bootstrap replicates, gamma-distributed substitution rates with invariant sites in five discrete categories (+G, parameter = 1.698), partial deletion with a cut off at 80%, a nearest-neighbor-interchange heuristic method, initial tree constructed automatically with NJ/BioNJ, branch swap filter very strong, number of threads 7. In total, 74 sequences with 434 alignment positions were used for the phylogenetic reconstruction. The resulting tree was exported in Newick format, including branch lengths and bootstrap values, and finally arranged using the iTOL (interactive Tree Of Life) server (https://itol.embl.de, accessed on 12 March 2023) [18].

## 3. Results and Discussion

### 3.1. Phylogeny

With the recently discovered CPD pathway able to synthesize heme *b* and the increasing numbers of microbial genomes from all taxa that have been sequenced, it is now possible to obtain a more comprehensive view of heme biosynthesis and its phylogenetic history. Only a few organisms lack orthologs of the heme biosynthesis pathway, mainly because of secondary gene loss [4]. This is the case for some bacterial pathogens and specific symbionts of domesticated organisms that obtain heme or intermediates from their host or other microorganisms living in the same niche, or for which heme availability is not essential. Analysis of the 981 available bacterial genomes revealed that 82% of *Actinobacteria*, 21% of the Firmicutes, 55% of *Chloroflexi*, and three of the four known *Acidobacteria* encode the genes of the CPD pathway, whereas *Proteobacteria*, except for the *Deltaproteobacteria*, and *Cyanobacteria* mainly encode genes from the PPD pathway [4,19]. For phylogenetic reconstruction, 74 porphyrinogen-type oxidase sequences from different prokaryotes and a limited number of eukaryotes (outgroup) were aligned and analyzed by the maximum likelihood method, using the optimal amino acid substitution model. Data mining is an important and critical step in the course of a phylogenetic study. Starting sequences of PgoX from *Myxococcus xanthus* and CgoX from *Bacillus subtilis* (CgoX) were used, as these sequences have already been studied on the purified protein level and can be considered to be reliable. Further, structural data from X-ray crystallographic studies are available for these two representatives. The *Basic Local Alignment Search Tool* by NCBI, or more specifically the Protein BLAST function, was fed with the respective amino acid sequences and thousands of hits were obtained. We have selected only reviewed sequences, because numerous misannotated sequences were detected. Misannotation can stem from contaminations of the original bacterial culture but sequencing errors can also eventually lead to flawed raw data, which is deposited in data banks. Nevertheless, even in reviewed sequences most often putative CgoX enzymes are referred to as PgoX. These inconsistencies arise from the relatively recent discovery of the CPD pathway. A major finding of this study is that, based on phylogeny, it becomes clear how to distinguish between PgoX and CgoX enzymes, and that this falls in line with the general differentiation between the protoporphyrin-dependent (PPD) heme biosynthesis and the coproporphyrin-dependent (CPD) heme biosynthesis pathways. 

The obtained phylogenetic tree (Figure 2) had high bootstrap support in all resolved clades and clearly depicted the main directions of this protein family’s evolution. At the origin of porphyrinogen-type oxidase evolution, PgoXs from numerous cyanobacteria have been found [20,21], which are distinctly separated from the descendant PgoX representatives of *Deltaproteobacteria* and from all later PgoX variants, leading to many eukaryotic clades. The position of cyanobacterial enzymes is unsurprising, as protoporphyrin IX (PPIX), an intermediate of the PPD pathway, is the precursor for chlorophyll biosynthesis, which was previously synthesized in certain ancient cyanobacteria lacking light-harvesting phycobilisomes [22]. Therefore, the cyanobacterial PgoX clade can be understood as a basal clade of the reconstructed tree, near the supposed ancient root of the whole PgoX/CgoX protein family. It is well documented that cyanobacteria are the first organisms known to produce molecular oxygen. By producing and releasing oxygen as a byproduct of photosynthesis, cyanobacteria are thought to have transformed the early, oxygen-poor, reducing atmosphere into an oxidizing one, causing the Great Oxidation Event and the “rusting of the Earth” that dramatically changed the composition of life forms on Earth [23].

By following further steps of this gene family’s evolution in the reconstructed evolutionary tree, it is clear that CgoX sequence variants originating from Gram-positive (monoderm) *Actinomycetota* and *Bacillota* are well separated from predecessor clades containing older PgoX types and that they represent a peculiar type of gene speciation (Figure 2). In the central part of the phylogenetic tree, colored gray blue, ocher, and blue, respectively, three main directions of this protein family evolution, which are supposed to be all CgoX representatives, can be observed. *Actinomycetota* CgoX representatives (gray-blue panel) were first segregated during the formation of CgoX variants. In this branch, an early diverging clade of pathogenic *Corynebacteria* is interesting. These bacteria are well-known, dangerous pathogens and can potentially obtain heme from their hosts; however, early in the evolution of this branch, they evolved a specific CPD pathway to biosynthesize this cofactor. Several non-pathogenic soil bacteria were found among the descendant clades of this branch. This finding supports the hypothesis that the evolution of CgoX preceded the pathogenic orientation of some bacteria.

*Thermomicrobiota*, *Nitrospirota*, *Chloroflexota*, and *Acidobacteriota* (Ocher panel) are clustered in a separately resolved branch. They are shown in the middle part of the reconstructed tree. The presented sequences originated from non-pathogenic bacteria in aquatic and terrestrial habitats, underscoring that they all require an efficient heme biosynthesis pathway. 

A distinct third branch of the *Bacillota* CgoX, with numerous well-separated clades, was also formed during this particular step of evolution and is shown in the lower part of this evolutionary direction in the dark blue panel. This represents a sister group of the *Thermomicrobiota* branch and is formed by both pathogenic and non-pathogenic bacterial species. This led to the conclusion that the corresponding CgoX variant evolved among *Bacillota* independent of their (potential) pathogenic orientation. It was already present in their probable non-pathogenic predecessor (shared with the *Thermomicrobiota*), which led to the diversification of all related *Bacillota* CgoX clades.

### 3.2. Sequence Alignment and 3D Structures

Eight porphyrinogen-type oxidase structures are currently deposited in the PDB: six PgoXs and two CgoXs. Of the six PgoX structures, two originated from the Gram-negative bacterium *Myxococcus xanthus* (2IVD, 2IVE) [24] with the inhibitor acifluorfen (AF) present as a ligand in one structure (2IVE). Three structures were from humans (3NKS, 4IVM, 4IVO) [25] and one was from tobacco (1SEZ) [14]. Of the two CgoX structures, the structure from *Bacillus subtilis* (3I6D) is the best characterized [26], whereas the structure from *Exiguobacterium sibiricum* (3LOV) has no associated publications. The overall structures of PgoX and CgoX are very similar, as both have a three-domain subunit architecture with a FAD-binding domain, a membrane-binding domain, and a substrate-binding domain. However, these are frequently referred to as pseudodomains because they do not fold independently [24]. The FAD-binding domain consists of residues from all three regions (Figure 3, residues highlighted in yellow): 12–66, 254–289, and 366–453 (*B. subtilis* numbering), which are conserved in both porphyrinogen oxidases. FAD does not covalently bind to these proteins. A recent study confirmed that D65 in *B. subtilis* coproporphyrinogen oxidase (*Bs*CgoX) is involved in a polar network crucial for enzymatic activity through the stabilization of the microenvironment of the isoalloxazine ring in FAD [27]. The active site is a hydrophobic cavity at the interface of the three domains. None of the resolved structures bound the substrate protoporphyrinogen IX or coproporphyrinogen III; however, AF was bound in the *Bs*CgoX and *M. xanthus* (*Mx*PgoX) structures, but in entirely different orientations. AF is a bicyclic, nonpolar molecule that mimics half of the protoporphyrinogen macrocycle. The AF molecule in *Bs*CgoX is located near the isoalloxazine ring of FAD and interacts with the residues of all three domains. Specifically, the benzene ring of the 2-nitrobenzoic acid moiety of AF is parallel to the isoalloxazine ring of FAD (Figure 4A), forming a pi-stacking interaction, while the 2-chloro-4-trifluoromethylphenoxy moiety is mostly exposed to the solvent [26]. In contrast, the binding site of the AF molecule in *Mx*PgoX was located deep inside the binding pocket and was accommodated by a hydrophobic pocket. 

Thus, the two individual AF molecules of both structures together form a macrocycle of protoporphyrinogen IX. A model of protoporphyrinogen IX oxidase (the coordinates for this model are not available in the PDB), which has two propionates less than the substrate coproporphyrinogen III complexed with *Bs*CgoX, shows that the side chain of I176 and the isoalloxazine ring of FAD form a sandwich around ring D of protoporphyrinogen IX. Rings A and C are likely stabilized by hydrophobic interactions with T366 and F227, respectively. The propionyl groups point towards the solvent region and may form ionic interactions with the side chain of K71. In the sequence alignment (Figure 3, blue), only the phenylalanine residue at position 227 (*B. subtilis* sequence numbering) was strictly conserved in CgoX. Leucine, valine, methionine, and glutamine occupy these positions in PgoX. The positions of K71 and Y366 were not conserved in CgoX or PgoX. Nevertheless, in CgoX from *B. subtilis*, the mutants I176A and K71A lead to a 30- and 50-fold decrease in catalytic efficiency, respectively, and the mutation Y366N results in the change of a hydrophobic side chain to a polar side chain, leading to a nearly 100-fold decrease in catalytic efficiency [28]. Another highly conserved residue in CgoX is R364, where the nitrogen (ἡ1) of the guanidinium group of arginine, which is 5 Å away from the nitrogen atom of the isoalloxazine ring, and the nitrogen ἡ2 point to the oxygen bridge between the 2-chloro-4-trifluoromethylphenoxy moiety and the 2-nitrobenzoic acid moiety of AF. The distance between the nitrogen ἡ2 and the oxygen is 5.1 Å. Mutation of this residue to leucine or aspartate in *Bs*CgoX reduced its catalytic efficiency by 38 and 3.3% [27]. In contrast, arginine was replaced by threonine in PgoX and serine in *Mx*PgoX (Figure 3).

The AF molecule and protoporphyrinogen IX in the model (the coordinates for the docked protoporphyrinogen IX are not available in the PDB) of *Mx*PgoX interact with R95 and F329 (Figure 3, *M. xanthus* numbering, residues are highlighted in brown), in addition to the shape of the pocket and water-mediated interactions with oxygen atoms. In the protoporphyrinogen IX-docked model of *Mx*PgoX, the tetrapyrrole ring is located between the isoalloxazine ring of FAD and the carbonyl oxygen of G167 [24]. G167 is conserved in both tetrapyrrole oxidases, and its carbonyl oxygen is likely used to center the tetrapyrrole macromolecule. On the opposite side of the tetrapyrrole ring, the width and bottom of the cavity were limited by R95, F329, G330, F331, L332, and I345 (Figure 4B). 

As shown in the sequence alignment (Figure 3), R95, F329, and L332 were strictly conserved in PgoX, whereas, in all CgoXs, they were always relatively small residues, such as serine, valine, or threonine. They appear to open the active site cavity, as discussed below. Only in the position of F329 in CgoX in the group of *Thermomicrobiota*, *Acidobacteria*, *Chlorophlexota*, and *Nitrospirota* bulky amino acids, such as arginine and phenylalanine, can this be found. Another interesting amino acid is N63, which is located on the same side as R95; F329 has been shown to be essential for activity and structural stability. It is equivalent to R59 in human PgoX and the R59W mutation is by far the most common mutation in variegate porphyria. Interestingly, only eukaryotic PgoXs have an arginine at this position, whereas most cyanobacteria and Delta-proteobacteria have an asparagine at this position.

As described above, the active sites of PgoX and CgoX differ in the size of their putative substrate cavities. The active cavity volumes of *Bs*CgoX, *Mx*PgoX, and human PgoX are shown by Qin et al. [21]. Surprisingly, the calculated volumes were 1173 Å^3^, 627 Å^3^, and 440 Å^3^ for *Bs*CgoX, *Mx*PgoX, and human PgoX, respectively, indicating that the active site cavity of *Bs*CgoX was almost 2- to 3-fold larger than that of *Mx*PgoX and human PgoX. They also calculated the potential distribution on the surface of the active site cavity; surprisingly, the surface of *Bs*CgoX was mostly positively charged, whereas the surfaces of the active site cavity in *Mx*PgoX and human PgoX were not. Coproporphyrinogen has two more negatively charged propionyl groups at positions 2 and 4, and is bulkier than protoporphyrinogen IX, which has two vinyl groups at these positions. Therefore, the larger substrate-binding cavity and positively charged surface of CgoX should favor the binding of coproporphyrinogen.

## 4. Conclusions

In this study, we have shown that, in the reconstructed maximum likelihood phylogenetic tree, gene sequences annotated for protoporphyrinogen oxidases have an evolutionary origin in cyanobacteria and are distinctly separated from sequence variants originating from the monoderm *Actinomycetota* and *Bacillota*, which are apparent CgoXs. Another important message of this work is that many CgoXs are still incorrectly annotated in public databases, as they were classified prior to the identification of the coproporphyrin-dependent (CPD) heme biosynthesis pathway. A clear distinction between these two enzyme classes is evidently given by the clustering of sequences in the phylogenetic analysis. This also clearly demonstrates that the evolution of CgoX occurred prior to the pathogenic orientation of some of these bacteria. The sequence alignment and structural comparison of PgoX and CgoX highlighted the similarities and differences between the two protein families. Further structural, biochemical, and biophysical studies on the physiological substrates of CgoX are necessary to determine the exact reaction mechanism at work. A substrate- or product-bound experimental structure would be a suitable starting point to further investigate the structure–function relationship of these enzymes.

## Figures and Tables

**Figure 1 biology-12-01527-f001:**
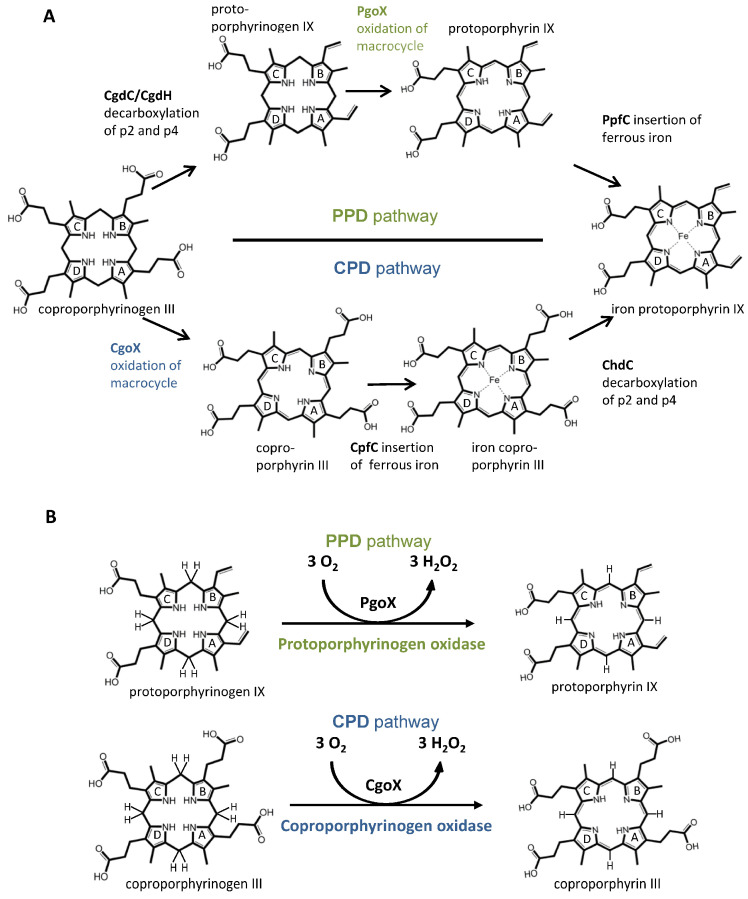
(**A**) Schematic overview of the coproporphyrin- and protoporphyrin-dependent pathways; (**B**) reactions that are catalyzed by protoporphyrinogen oxidase (PgoX, green labels) and coproporphyrinogen oxidase (CgoX, blue labels).

**Figure 2 biology-12-01527-f002:**
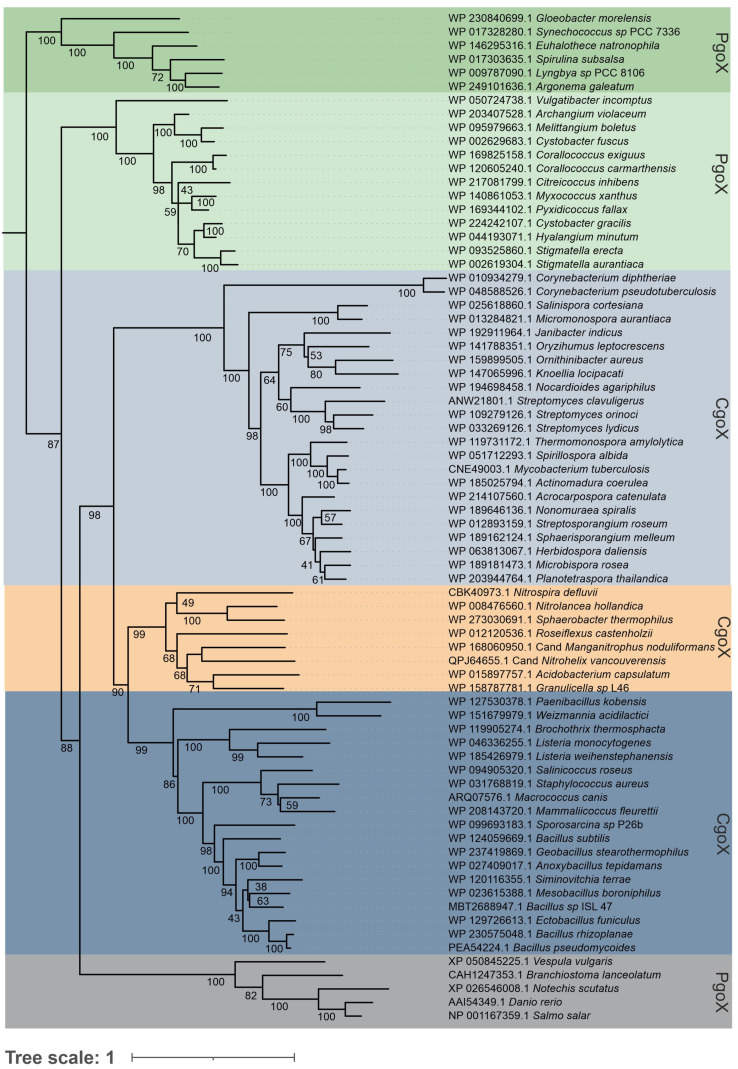
Reconstructed phylogenetic tree of a selection of 74 CgoX and PgoX sequences that was collected from public databases (Uniprot, NCBI). Bootstrap values are presented based on the obtained maximum likelihood (ML) output. Color code: green, *Cyanobacteria*; light green, *Deltaproteobacteria*; gray blue, *Actinomycetota*; blue, *Bacillota*; ocher, representatives from *Thermomicrobiota*, *Acidobacteria*, *Chlorophlexota* and *Nitrospirota*; gray, *Eukaryota.* Tree scale means an internal measure for the frequency of observed amino acid substitutions.

**Figure 3 biology-12-01527-f003:**
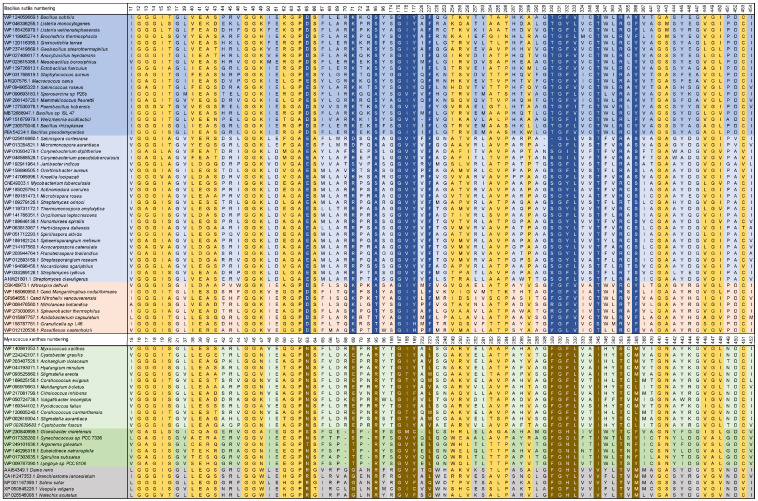
Selected parts of the multiple sequence alignment of 74 CgoX and PgoX sequences. Colour code: green, *Cyanobacteria*; light green, *Deltaproteobacteria*; gray blue, *Actinomycetota*; blue, *Bacillota*; ocher, representatives from *Thermomicrobiota*, *Acidobacteria*, *Chlorophlexota* and *Nitrospirota*; gray, *Eukaryota*; yellow boxes, FAD-binding residues; blue boxes, highly conserved residues in CgoX or residues that interact with AF and the modeled protoporphyrinogen IX (numbering according *Bs*CgoX); brown boxes, highly conserved residues in PgoX or residues that interact with AF and protoporphyrinogen IX in the docking experiment (numbering according *Mx*PgoX).

**Figure 4 biology-12-01527-f004:**
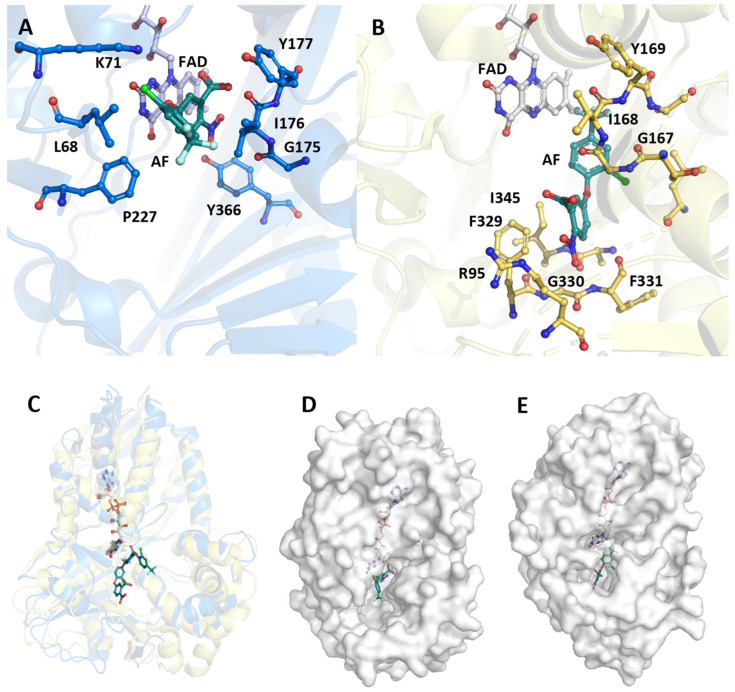
Comparison of the structure of *Bs*CgoX (3I6D) and *Mx*PgoX (2IVD). (**A**) Active site structure of *Bs*CgoX (carbon atoms are depicted in blue) in complex with AF (turquois); (**B**) active site structure of *Mx*PgoX (yellow) in complex with AF (turquois)—the FAD is presented by grey carbon atoms, oxygen atoms are shown in red, and nitrogen atoms in blue; (**C**) overlay of the monomeric structure of *Bs*CgoX (light blue) and *Mx*PgoX (light yellow) in complex with AF (turquoise green); (**D**,**E**) access to the active site cavity of *Bs*CgoX and *Mx*PgoX.

## Data Availability

All data used in this study were from public databases. Sequences were from UniProt and NCBI, and protein structures were from the PDB database.

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
