# Peer review of "The Molecular Evolution, Structure, and Function of Coproporphyrinogen Oxidase and Protoporphyrinogen Oxidase in Prokaryotes"

_biology, 2023, doi:10.3390/biology12121527_

Round 1

Reviewer 1 Report

Comments and Suggestions for Authors

Dear editor

The study conducted by Zamocky et al., performed a comprehensive analysis of Coproporphy-rinogen Oxidase and Protoporphyrinogen Oxidase in some Prokaryotes. The approach of the authors for the development and relevance of the subject is adequate, the information provided is orderly and coherent but not too much to be published.  Such studies could be done by student and not by scientists with high level. I think that those fundings are still preliminary. I suggest to add some other analysis

1-     The protein phosphorylation is crucial for protein. Could you pleas eidentifiy how many phosphorylation reesidues in those proteins ?

2-     Identify the Promoter Cis-Regulatory Element Analysis

3-     Analyses of Genes Structures and Conserved Motifs

Reviewer 2 Report

Comments and Suggestions for Authors

I think that the manuscript entitled “Molecular Evolution, Structure, and Function of Coproporphyrinogen Oxidase and Protoporphyrinogen Oxidase in Prokaryotes” is in principle suited for a publication in Biology, Special Issue “Approaches for the Reconstruction of Protein Families’ and Superfamilies’ Evolution with Consequences for De Novo Protein Design 2.0”. Overall, the research topic is relevant and could be of interest to a wide readership in the journal. However, I have significant concerns and doubts regarding the choice of objects for phylogenetic analysis and the research methodology.

Comments:

Line 67. It would be clearer to mention that Figure 1 is schematic in the caption.

Lines 99-100. In "PgoX is a membrane-bound dimer, while CgoX is a soluble monomer," it might be helpful to provide a brief explanation of what a "membrane-bound dimer" and "soluble monomer" mean in this context.

Lines 118-119. Please, in Table S1, specify which sequences were obtained from NCBI and which ones from UniProt. Additionally, in Table S1, it would be desirable to indicate which ones are CgoX and which ones are PgoX.

Lines 135-140. Why were these specific parameters chosen?

Line 189. “In the central part of the phylogenetic tree, colored gray-blue, ocher, and blue, respectively, three main directions of this protein family evolution, which are supposed to be all CgoX representatives, can be observed”. Could you please explain why, according to NCBI data, the sequence in Table S1 as >WP_119731172.1_Thermomonospora_amylolytica_Actinomycetota is labeled as protoporphyrinogen oxidase, while in Figure 2, this sequence clusters and is labeled as coproporphyrinogen oxidase (CgoX)? The same for >WP_013284821.1_Micromonospora_aurantiaca_Actinomycetota, >WP_127530378.1_Paenibacillus_kobensis_Bacillota, and many others. At the same time, NCBI contains many other protein sequences corresponding to coproporphyrinogen III oxidase (for example, https://www.ncbi.nlm.nih.gov/protein/PNS08567.1, https://www.ncbi.nlm.nih.gov/protein/1443518651, and others) that were not included in the study. Why were these specific only 74 sequences chosen? I think the authors should have, for comparison (as a control), conducted a phylogenetic analysis only of those coproporphyrinogen oxidases for which the mechanism of action is confirmed and there is no confusion in their designation in NCBI as protoporphyrinogen oxidases.

Reviewer 3 Report

Comments and Suggestions for Authors

In this paper, the authors present the phylogenetic and structural relationships of CgoX and PgoX in the heme biosynthesis. With respect to the subject matter, I think that it is an interesting topic and can be a starting point for future studies and analyses. 

The structural part of the paper seems clear and well structured. The same happens with the images, which are very illustrative and clarifying, giving support to what is expressed in the text.

The same is not true for the initial part of the paper (the abstrat and part of the introduction). Although the theoretical description is well presented and easy to understand, I think that the motivation for the work and the objectives should be expressed much more clearly. Emphasis should be placed on the reasons that lead the authors to carry out this study and which are the objectives they want to achieve. It would also be interesting to go more deeply into what advantages or contributions this study will provide for future work.

Something similar happens with the conclusion. The authors present it very briefly, without offering a robust final approach or elaborating on the implications of the conclusions they have drawn.

Given that the methodological part seems well presented and the work itself is interesting, I recommend that the paper be accepted, but with modifications to a part of the introduction and the conclusion. The changes I propose aim to focus directly on the objectives and purpose of the work, making it easier for the user to read the paper.

In a more concrete way:

On line 51 - Replace . by , so as not to split the sentence.

On line 118 - What criteria are used to select the sequences? And why this number of sequences? It should be explained on what basis they selected that number and type of data.

On line 134- The authors refer to Le Gascuel model of amino acid substitutions [18] as the model with the lowest BIC value. How did they obtain this information? What software did they use for this?

On line 150-151: "Only a few organisms lack orthologs of the heme biosynthesis pathway, mainly because of secondary gene loss". This sentence is not clear enough. It would be interesting to explain it a bit more.

Round 2

Reviewer 1 Report

Comments and Suggestions for Authors

Thank you for considering my comments. it has improved the quality of the manuscrit 

Author Response

Reviewer #2

Comments and Suggestions for Authors:

Thank you for considering my comments. it has improved the quality of the manuscript.

Answer: we thank this reviewer for his positive response.

Reviewer 2 Report

Comments and Suggestions for Authors

The authors have addressed most of the comments and made revisions to the manuscript and supplementary material. It is slightly concerning that the response states: "It is not possible to perform a phylogenetic analysis of only those coproporphyrinogen oxidases for which the mechanism of action is confirmed, because there are only a few CgoX, such as for Bacillus subtilis, for which the reaction mechanism is confirmed”. Yet, it is precisely these coproporphyrinogen oxidases that would be of the greatest interest and utility for such an analysis. Nonetheless, the authors' work may still interest the readers of the journal Biology and prove useful for comparison in similar studies.

Minor comments:

Please indicate what "Tree scale: 1" means in Figure 2.

Line 209: Please clarify the abbreviation "ML," as it seems to be the first instance of its use.

Author Response

Reviewer #3

Comments and Suggestions for Authors:

The authors have addressed most of the comments and made revisions to the manuscript and supplementary material. It is slightly concerning that the response states: "It is not possible to perform a phylogenetic analysis of only those coproporphyrinogen oxidases for which the mechanism of action is confirmed, because there are only a few CgoX, such as for Bacillus subtilis, for which the reaction mechanism is confirmed”. Yet, it is precisely these coproporphyrinogen oxidases that would be of the greatest interest and utility for such an analysis. Nonetheless, the authors' work may still interest the readers of the journal Biology and prove useful for comparison in similar studies.

Minor comments:

Please indicate what "Tree scale: 1" means in Figure 2.

Answer: The meaning of a tree scale was now explained in the updated legend to Figure 2. It is an internal measure for the frequency of amino acid substitutions in aligned protein sequences.

Line 209: Please clarify the abbreviation "ML," as it seems to be the first instance of its use.

Answer: We have now explained the meaning of „ML“ in the legend to this figure but we have added this abbreviation already in the corresponding methods section at line 145 of the second revised version.